# Cooperation-Assisted Spectrum Handover Mechanism in Vehicular Ad Hoc Networks

**Ming-Chin Chuang**

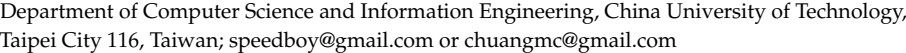

Department of Computer Science and Information Engineering, China University of Technology, Taipei City 116, Taiwan; speedboy@gmail.com or chuangmc@gmail.com

**Abstract:** This paper proposes a cooperation-assisted spectrum handover (CASH) scheme in vehicular ad hoc networks (VANETs). In CASH, each vehicle uses cognitive radio technology to collect the surrounding spectrum information, computes the stability of the spectrum, exchanges neighbor information by vehicle-to-vehicle collaboration technology, and then executes a partial prescan mechanism to reduce the handover delay. The proposed method can improve the spectrum utilization, shorten the delay time of the spectrum handover, and reduce the total number of handovers. Finally, the simulation results show that the proposed method outperforms other existing schemes. In spectrum utilization, CASH is about 20% better than that of the traditional full scan method. In average scan time, the time of CASH is about 5 times less than the time of the traditional full scan method. In total number of handovers, CASH improves the performance of the full scan scheme by 33%.

**Keywords:** cooperation-assisted; spectrum handover; vehicular ad hoc network; prescan





## 1. Introduction

Due to the rapid development of wireless communication technology, a variety of application services have been proposed. Vehicular ad hoc networks (VANETs) are classified as an application of mobile ad hoc networks (MANETs) [1] that have the potential to improve road safety and to provide comfort to travelers. VANET has the following unique characteristics [2]: predictable mobility, providing safe driving, improving passenger comfort and enhancing traffic efficiency, no power constraints, variable network density, rapid changes in network topology, large scale network, and high computational ability. Recently, the Internet of Vehicles (IoV) has received much attention from industry, government, academics, and research units. Unfortunately, there are still many application services encountering bottlenecks in VANETs due to the bandwidth limitations of the wireless spectrum. Therefore, it calls for increasing the utilization rate of the spectrum to satisfy these services as much as possible. Although cognitive radio technology has been proposed to improve spectrum utilization, there are still many issues that need to be addressed. For example, when a vehicle hands over the base stations (BSs), the issue of spectrum handover needs to be considered. The network performance gets worse as the vehicle moves faster.

2019 is the first year of 5G. Various telecom operators spared no expense to obtain a concentrated and relatively large spectrum in the 5G spectrum, which led to a record high in bidding amount. In other words, this also highlights the high value of the wireless spectrum. The significant features of IMT-2020 (also called 5G) [3] include enhanced mobile broadband (eMBB), ultra-reliable and low latency communications (URLCC), and massive machine-type communications (mMTC). Obviously, 5G technology will play an important role in the IoV environment.

According to the statistical results of the Federal Communications Commission in the United States (FCC) [4], the spectrum utilization efficiency of 0 to 6 GHz is about 15% to 85%. The results mean that the part of the frequency spectrum is overcrowded, but the part

of the frequency spectrum shows a sharp contrast that is rarely used. It can be seen that the existing method of using fixed spectrum allocation will result in very uneven spectrum utilization. Therefore, if the wireless communication equipment of the vehicle is capable of detecting the use of the local spectrum and can make full use of the idle spectrums, it will improve the overall network performance and maximize the benefits of the spectrum. In the vehicular network, the issue of the spectrum handover needs to be considered when a vehicle hands over the BSs. The network performance gets worse as the vehicle moves faster. Given this, we will propose a novel spectrum handover method based on cognitive and cooperative techniques, called cooperation-assisted spectrum handover (CASH). The detailed operations of CASH will be described in the following sections.

The remainder of this paper is organized as follows. Section 2 provides a review of related work. In Section 3, we describe the proposed spectrum handover scheme and in Section 4 we show the performance results. Section 5 contains some concluding remarks.

## 2. Related Work

### 2.1. IEEE 802.22

Global wireless spectrum resources are gradually in short supply, and national telecommunication spectrum management agencies are actively discussing how to effectively use spectrum. A perceived wireless network is an intelligent wireless communication system that can self-detect the surrounding wireless environment to adjust the operating parameters of its wireless devices in real time to adapt to changes in the external wireless environment. Figure 1 shows the flow chart of the cognitive radio technology.

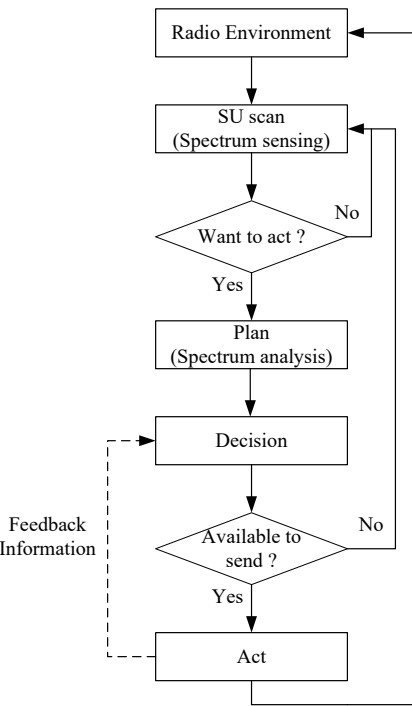

**Figure 1.** The flow chart of the cognitive radio technology.

The Institute of Electrical and Electronics Engineers (IEEE) established an 802.22 working group [5] in October 2004. Its main task is to develop and establish a set of wireless wide area network standards based on perceptual wireless network technology. This technology allows secondary users (SUs) to use idle TV broadcast spectrums for communication without affecting the primary user (PU). The same area may be covered by the signal range of multiple BSs. To avoid mutual interference, IEEE 802.22 requires SUs to be equipped with at least two antennas, one of which is a directional antenna to reduce interference when uploading, and the other is an omnidirectional antenna to detect

spectrums that is not currently used by the PU. To dynamically use the idle spectrum, both the BS and the SU must be able to detect whether there is a PU. The BS scans the spectrums that may be idle and determines which spectrums are truly idle. Finally, the BS will select a spectrum from the idle spectrums as the operating spectrum, and select a spectrum as the backup spectrum. The operating spectrum is the spectrum that the BS mainly provides services, and the backup spectrum is the spectrum that the SU will switch to when a PU appears. The BS will periodically broadcast these messages to all SUs in the signal range of the BS. Mohammad et al. [6] proposed an enhancing spectrum utilization scheme in cellular mobile networks by using cognitive radio technology. However, this scheme does not take the mobility situation into account.

### 2.2. Existing Spectrum Handover Schemes

Previous studies [7–10] proposed spectrum management schemes in railway networks. These schemes provide significant improvement to network performance due to their knowledge of the the trajectory and speed of the train. However, the trajectory and speed of each vehicle are changeable. This will cause difficulty in spectrum management. Many spectrum handover methods [11–16] assume that the SU can accurately detect the conditions of all spectrums in order to find a suitable spectrum for handover. However, this assumption is too strong and is unlikely to happen in reality. Moreover, these schemes do not take into account the movement of users. In fact, the SU in the vehicle network will need to spend more time scanning the spectrum to avoid affecting the performance of the PU. Figure 2 shows the operations of the traditional spectrum handover. We can see that the SU needs to spend more time scanning the spectrum when the number of BSs increases.

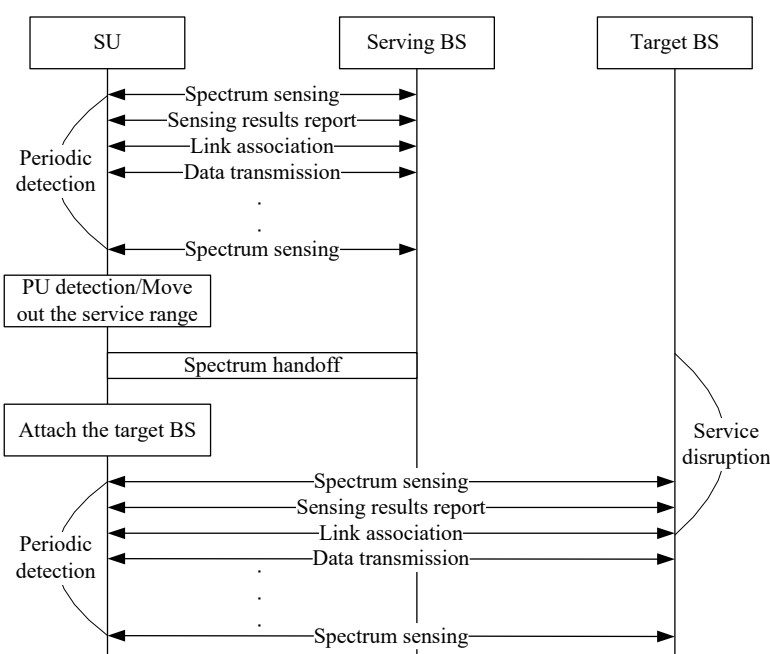

**Figure 2.** The traditional spectrum handover.

### 2.3. Cooperative Assistance Delivery Mechanism

In recent years, cooperative communication technology is applied for improving the transmission performance. The cooperative communication has two modes, one is the relay mode, and another is the co-transmission mode, as shown in Figure 3. The relay mode is mainly because the relay point is located between the transmitting end and the receiving end, so the relay point can use a relatively high transmission rate to help transmit data and reduce the overall transmission time. In the co-transmission mode, the cooperating node and the transmitting end form a virtual cluster through synchronization, and then transmit the same piece of data with less power, to save power. There have been

many documents [17–20] using these two methods to improve the performance of network transmission. In terms of the mobility management mechanism, the studies [21–24] propose the cooperative mechanisms at the network layer. These mechanisms help mobile devices obtain new network IP addresses in advance through relay node. The most time-consuming process is to obtain the IP address in the network layer, because the IP address must be unique and cannot be the same as that of other people. Chang et al. [25] proposed a group-based mobility management scheme in VANETs, but they do not consider the spectrum handover issue.

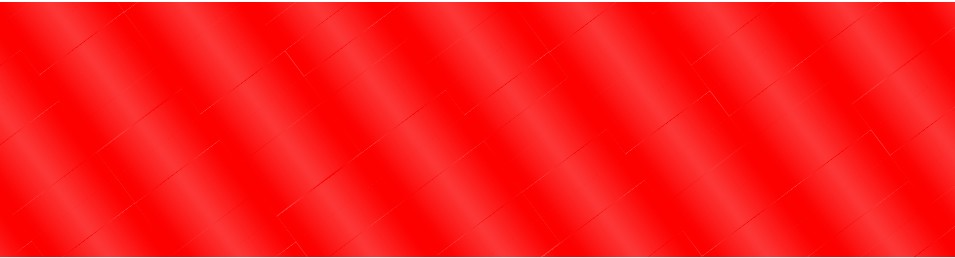

**Figure 3.** Cooperative transmission: (**a**) relay mode and (**b**) co-transmission mode.

### 3. Proposed Scheme

This section will describe our cooperation-assisted spectrum handover (CASH) scheme in detail. Figure 4 shows the concept of the cooperation-assisted spectrum handover. PU is the primary vehicle and SU is the secondary vehicle. Each PU communicates with a BS and other SUs at the same time because it is equipped with multiple antennas. In addition, each vehicle can obtain the spectrum information of neighboring BSs in advance through multi-hop V2V communications; it then makes the best handover decision. When the PU or SU hands over the new BS, it directly switches to the selected spectrum without scanning the overall spectrum. Therefore, the CASH scheme can save the scanning time and mitigate the impact of switching spectrum (e.g., packet loss and performance degradation).

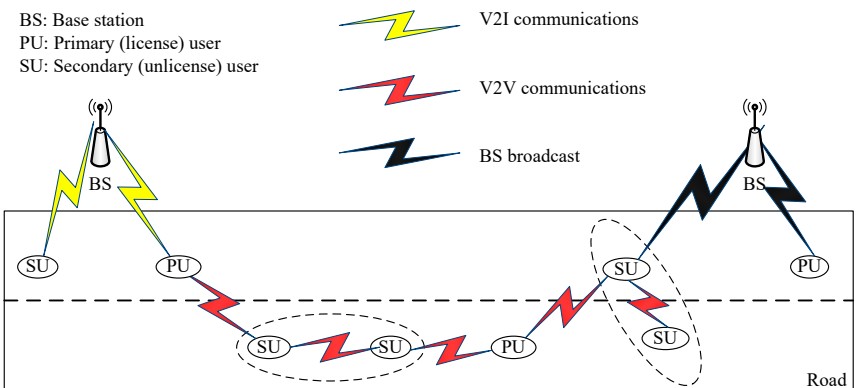

**Figure 4.** The concept of the cooperation-assisted spectrum handover (CASH).

### 3.1. Assumptions

In this paper, the proposed scheme is based on the above two assumptions.

(1) Assumption 1: In this study, we assume that adjacent spectrums will not interfere with each other. In fact, many technologies can mitigate the interference problem, such as inter-cell interference coordination (ICIC) and interference cancellation technology.

(2) Assumption 2: According to the IEEE 802.22, the BS will regularly broadcast to inform the local area of the spectrum usage status. Therefore, we assume that all SUs can get the immediate spectrum usage status. Our partial scanning method is based on this information to select the suitable channels for scanning. If the SU has not entered the signal range of the new BS, it is still possible to obtain spectrum information through the multi-hop V2V communication mode.

### 3.2. Partial Spectrum Scan Scheme

This study proposes a rapid spectrum scanning mechanism, which relies primarily on the IEEE 802.22 standard agreed to make improvements. First, the BS regularly broadcasts the current usage status of all spectrums, and then PU and SU will receive this message, and then PU and SU will choose to use the active (proactive) or passive (reactive) spectrum scanning method to decide which spectrum to switch. The current scanning method, as shown in Figure 5a, can update the most recent spectrum information at any time. If the optimal spectrum changes, SU will perform spectrum handover. In contrast, the passive scanning method is shown in Figure 5c. If the spectrum quality is lower than a threshold, then it will start the process of scanning the spectrum. The spectrum quality is usually based on the strength of the received signal strength (RSSI), idle time, and the available bandwidth or the packet error rate (PER). The traditional method is to scan all spectrums (full scan), but later some scholars [26] proposed a partial scan method to shorten the scan time.

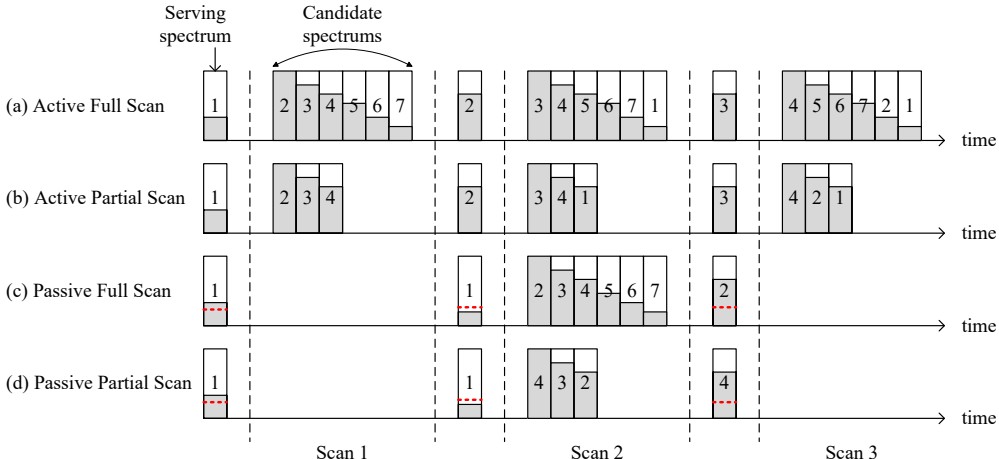

**Figure 5.** An example of a fast spectrum scanning mechanism.

The fast spectrum scanning mechanisms of this paper are based on active partial scanning and passive partial scanning. If the spectrum changes greatly, the active scanning method may cause constant switching of spectrums; if the spectrum change is small, the passive scanning method may not maximize the spectrum performance. Therefore, we can decide which scanning method to use, according to the stability of the spectrum change. Figure 5 shows an example of a fast spectrum scanning mechanism. In Figure 5b,d, we can see that using partial spectrum scanning can indeed reduce the scanning time and increase the data transmission time after a successful connection, thereby improving the overall network performance. In Figure 5b, the connection results are different each time (that is, $1 \rightarrow 2 \rightarrow 3$) because the SU selects the spectrum with the best quality to connect each time. As shown in Figure 5d, when the resource of the spectrum is lower than the threshold (the red dotted line in the figure), the scanning procedure will be performed to switch from spectrum 1 to spectrum 4.

### 3.3. Cooperation-Assisted Scheme

The cooperative assistance spectrum handover mechanism is divided into two parts, the first is the cooperation-assisted scheme combined with the fast scanning mechanism of the previous section, and the second is the discussion of spectrum stability. These two items will play an important role in the overall perception of the overall performance of the wireless system. The first part is mainly to shorten the time to scan the channel, and the second item is mainly to ensure the stability of the scan results. If the stability of the spectrum is too poor, it means that this spectrum is often occupied, but the length of the occupied time is uncertain. In this way, if the SU switches to this spectrum, it is very likely that some PU will use the spectrum soon, causing the SU to search for a new available

spectrum again. Due to the high-speed mobile nature of in-vehicle networks, stability is more of a challenge.

(1)    Cooperate to assist with the rapid scanning mechanism

The cooperative communication mode of this paper will adopt the relay method. Figure 6 shows the cooperation-assisted mechanism and cooperation trigger timing. In Figure 6a, SU 2 receives the current spectrum usage of the BS 2 broadcast, and then informs SU 1 through V2V communication to achieve a cooperative coordination mechanism. In this way, the SU 1 can obtain the usage status of the BS 2 spectrum before it has entered the range of the new BS, and then make early decision-making preparations. Figure 6b shows the circumstances under which SU 1 triggers the request for cooperation. Basically, if the RSSI of SU 1 to BS 1 is lower than the threshold, it will start to send out cooperation request messages. Once other SUs are in the signal range of different BSs, they will reply to SU 1, and provide their current BS spectrum usage status (e.g., available bandwidth or spectrum utilization rate). Figure 7 is a step-by-step diagram of cooperation-assisted mechanism combined with the fast scanning mechanism.

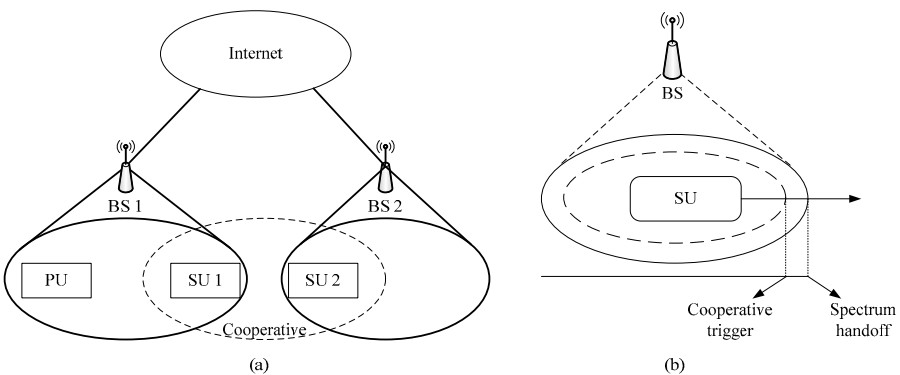

**Figure 6.** (**a**) Cooperation-assisted mechanism and (**b**) cooperation trigger timing.

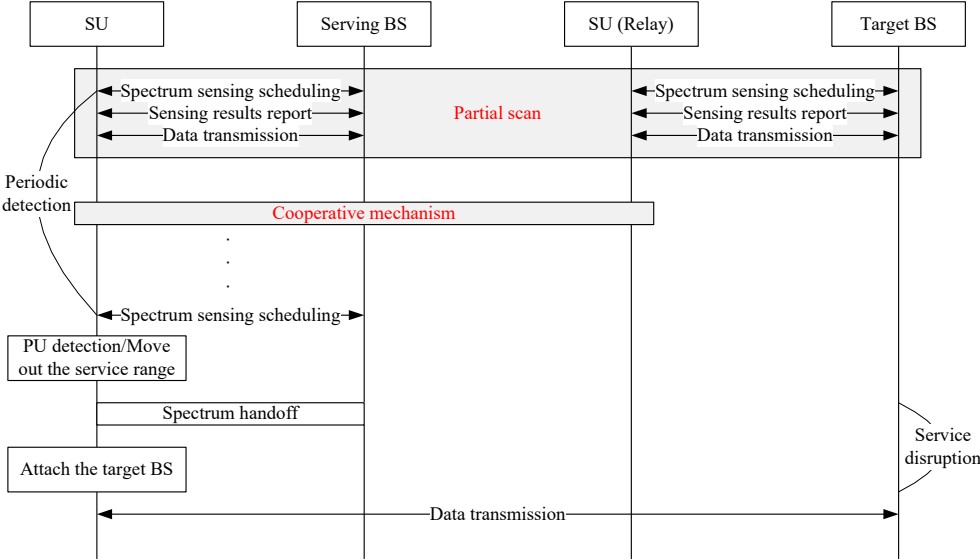

**Figure 7.** Cooperation-assisted with fast scan mechanism.

(2)    Spectrum stability judgment

If the variability of the spectrum resources is too large, we think that the stability of the spectrum is not good. In principle, we hope that the spectrum that the SU will switch to can meet the resource requirements of the SU and use the spectrum as long as

possible. As mentioned earlier, in order to avoid the ping-pong effect causing the SU to always perform handover behavior, we will use Equation (1) to determine the stability of each spectrum (that is,, stable index, SI). This index represents the degree of change in the available resources of the spectrum. The greater the change in the surrounding environment, the frequent execution of scanning procedures by the PU or SU will increase the number of potential handovers. Therefore, the greater the value of the SI, the poorer the stability of the spectrum. The term $\mu$ is the average of the available resource, $y_i$ is the available resources for each scan of the spectrum, $i=1, \ldots, n$, $N$ is the total of the scan times. Figure 8 shows an example of scanning three rounds.

$$\text{SI} = \frac{(y_1 - \mu)^2 + (y_2 - \mu)^2 + \cdots + (y_n - \mu)^2}{N} = \frac{\sum_{i=1}^{n}(y_i - \mu)^2}{N} \tag{1}$$

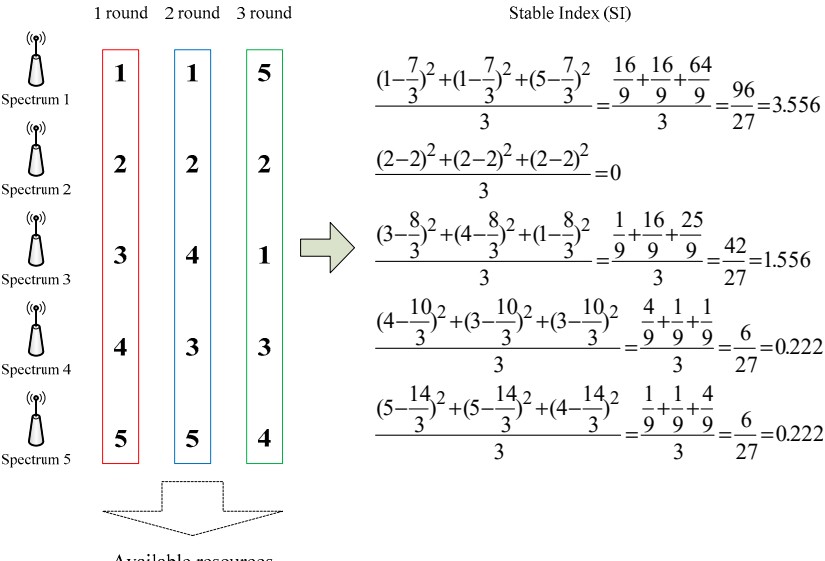

**Figure 8.** An example of scanning three rounds.

From the results presented in the Figure 8, we discuss three cases.

- Best case: Spectrum 2 has the highest stability because the variability of the spectrum is zero. Although Spectrum 2 is the most stable, it does not have the most resources available.
- General case: The SI values calculated for spectrum 4 and spectrum 5 are the same, which means that they have the same stability. If the stability is the same, we can choose a spectrum with more available resources to switch.
- Worst case: Spectrum 1 has the lowest stability, which means that the SU switches to this spectrum. However, radio resources will suddenly become inadequate, or a PU may suddenly want to use this spectrum. If the SU performs the handover procedure too frequently, it may not be able to effectively use the spectrum resources. If the SI values of all spectrums are high, we recommend using passive scanning methods to reduce the ping-pong effect. Finally, Figure 9 shows a flow chart of the cooperative assistance spectrum handover mechanism.

(3)   Dynamic scan period

In this section we will discuss dynamically adjusting the scan period due to the performance; the scan time is a trade-off. In Figure 10, we assume that a transmission period is M time slots and the sensing time takes m time slots. It means that the time that the SU can transmit is (M-m) time slots. If the sensing time is too short, it may not be able to scan all available spectrums; if the sensing time is too long, it may be compressed to

the transmission time, resulting in a decrease in overall spectrum performance. Therefore, we define Equation (2) to dynamically adjust the sensing time, so that the transmission performance can meet the system requirements. According to the 3GPP specification [27], the scan period has an actual upper limit, such as the neighbor cell list (NCL), which can only record up to 32 data. Therefore, we preset a complete scan time upper limit as the time to scan 32 spectrums. If we use partial spectrum scanning, it means that the system can complete several rounds of partial scanning within a complete scanning time. T represents the total time allocated to SU, $T_{tran}$ is the total transmission time, $T_{scan}$ is the total scan time, $P_{collision}$ is the probability of a SU collision during transmission, $P_{Fail}$ is the probability of transmission failure, and $T_{round}$ is the time consumption of scanning a spectrum. In order to scan multiple spectrums, $\alpha$ is the number of spectrums to be scanned partially, $\alpha$ must be less than or equal to 32, $\beta$ is the number of rounds to be scanned, and D($\alpha$, $\beta$) is the allocation ratio of the transmission time and the sensing time that the system wants. Figure 11 illustrates the effect of $\beta$ on the probability of SU transmission failure in the full scan mode. From the figure, we can observe two points. First, the collision probability is higher, and the SU transmission failure probability is higher. The second point is that the more scan rounds $\beta$, the probability of SU transmission failure is higher. The reason is that the more scans, the longer the sensing time, and the probability of the SU suffering a collision is higher. But the more rounds of scanning, SU can find a more stable spectrum to use.

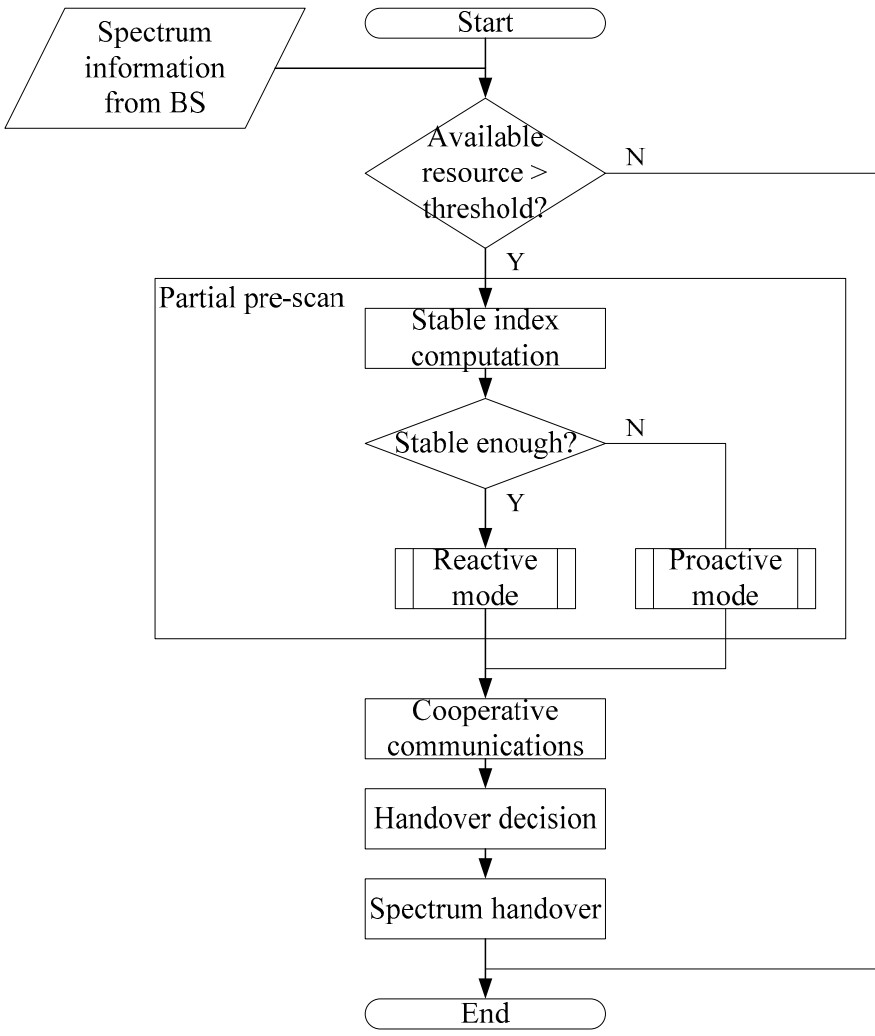

**Figure 9.** The flow chart of CASH.

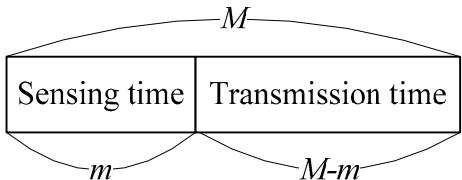

**Figure 10.** Time distribution diagram of transmission cycle.

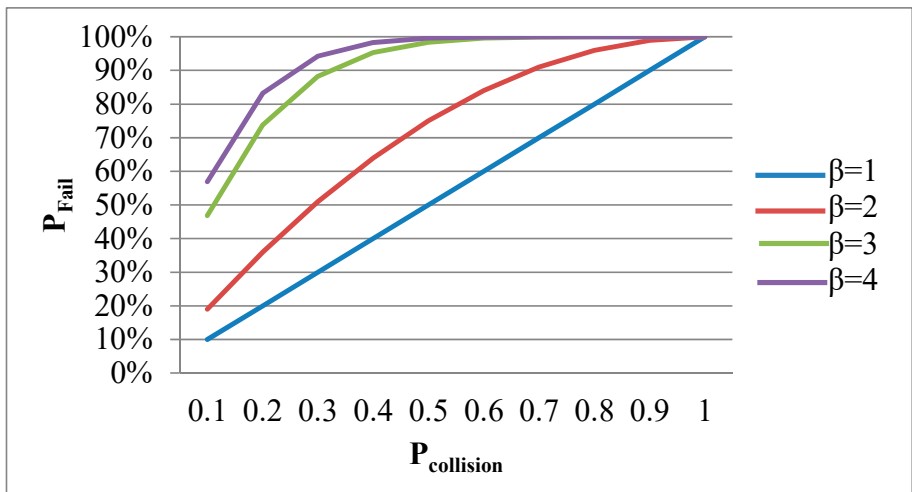

**Figure 11.** The influence of β on the probability of secondary user (SU) transmission failure in full scan mode.

Figure 12 shows the effect of different scanning modes on the transmission magnification within a fixed sensing time. The total sensing time is 32 time slots. D(32,1) is a full scan mode, and the others are partial scan modes. In Figure 12, we can observe three things: First, when the transmission failure probability is higher, the transmission time multiplier of the system is smaller. Second, when using the full scan method, a higher transmission time magnification can be obtained, but the stability is not necessarily better. Third, if the system would like to obtain a specific transmission time magnification, it can select an appropriate scan mode according to the current transmission failure probability. For example, if we want to obtain an 8 times transmission magnification, when the transmission failure probability is 10%, use D(2,16) scan mode; when the transmission failure rate is 30%, the D(8,4) scan mode is adopted; and when the transmission failure rate is 50%, the D(16,2) scan mode is adopted. As a result, if we want higher performance, we need to reduce the probability of transmission failure ($P_{Fail}$) and reduce the scan time ($T_{scan}$).

$$D(\alpha, \beta) = \frac{T_{tran}}{T_{scan}} = \frac{(T - T_{scan}) \times (1 - P_{Fail})}{\alpha \times \beta \times T_{round}} \tag{2}$$

$$P_{Fail} = 1 - (1 - P_{collision})^{\beta} \tag{3}$$

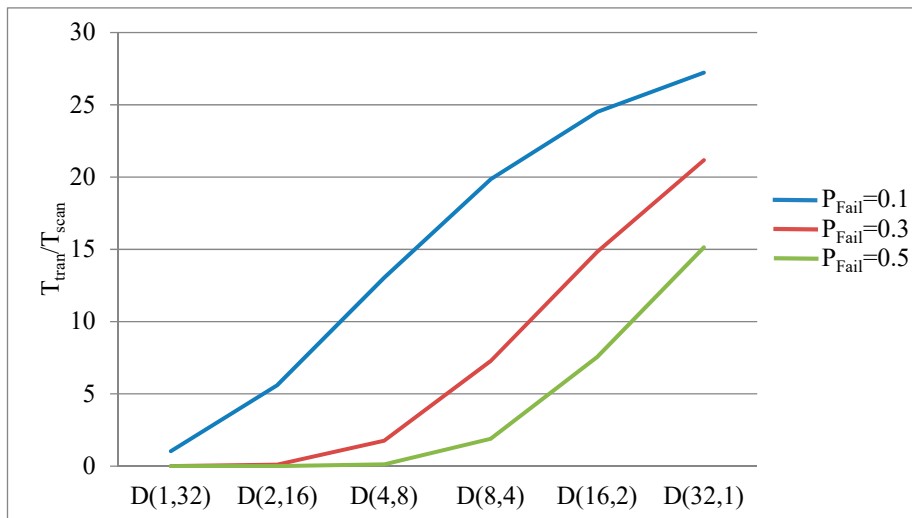

**Figure 12.** The influence of different scanning modes on the transmission magnification within a fixed sensing time.

Basically, we recommend using passive handover mode to avoid the ping-pong effect. Because the passive handover mode does not perform scanning and handover as long as the available resources are not lower than the threshold. If a partial passive scanning scheme is adopted, the scanning time and sensing time cost will be much less than the full scanning method. Moreover, the SU can select a spectrum with small variability (that is, SI) and meet demand for connection. Finally, we hope that the system can find a balance between transmission time and sensing time.

Actually, a collision-free SU is also an important issue in VANETs. If the traffic system can control the SU of each area, the probability of transmission failure will be improved. However, this topic has exceeded the scope of this paper. Therefore, we provide relevant literature to interested readers. Ashwin et al. [28] proposed a robot motion planning scheme using adaptive hybrid sampling in probabilistic roadmaps to get better performance. Mehdi et al. [29] proposed a near-optimal solution for a robotic automated storage and retrieval system (AS/RS). This solution included finding the optimal sequence of orders and optimal sequence of multi-location items inside each order that jointly minimized total travel times.

## 4. Performance Evaluation

This section contains three parts. The first part introduces the performance metrics, the second part describes the setting of simulation parameters and simulation environment, and the last part shows the simulation results. In addition, we will compare the performance of the following five spectrum handover methods.

(1) Full scan without the cooperation ($SCAN_{Full}$): When the SU enters a new BS, it starts to perform the full spectrum scanning method, and then selects the most idle spectrum to transmit.

(2) Random access without the cooperation ($SCAN_{Random}$): When the SU enters a new BS, it does not perform scanning and directly uses the random access spectrum to transmit data.

(3) Partial scan without the cooperation ($SCAN_{Partial}$): When SU enters a new BS, it adopts the partial spectrum scanning method to shorten the scanning time, and then the SU selects the idlest spectrum to transmit data according to the partial results of the scanning.

(4) Full scan with the cooperation ($SCAN_{Co-Full}$): The cooperating SU will collect the usage status of all the spectrums of the new BS. Therefore, before entering a new BS, the SU can know the spectrum usage status of the new BS in advance through the

V2V message exchange. The BS can quickly decide which spectrum to use to transmit data.

(5)  Partial scan with the cooperation ($SCAN_{Co\text{-}Partial}$): The cooperating SU will collect the usage status of some spectrums of the new BS. Therefore, before entering a new BS, the SU can know in advance the usage status of the new BS's spectrum through the V2V message exchange. The BS can quickly decide which spectrum to use to transmit data.

### 4.1. Performance Metrics

(1)  Spectrum utilization: The higher the spectrum usage, the better the design of the wireless cognitive sensing mechanism, and the higher the overall network performance.
(2)  Average scan time: The shorter the scan time, the shorter the interruption time of transmission and the increase of the time for transmitting data.
(3)  Total number of handovers: The handover situation may be due to the switching of the BS, the sudden use of the spectrum or the transmission collision of the SU. If the total number of spectrum handovers is too many, it means that the stability of the spectrum scanning mechanism is not good.

### 4.2. Simulation Parameter Settings

The simulation tool is the NS-2 [30]. Table 1 shows the setting of the simulation parameters. The simulation area is 2000 × 2000 m, the signal range of each car is 100 m, and the signal range of the BS is 1000 m, PU and the number of SU cars is 2 to 10, respectively, and the speed of the cars is 5 to 20 m/s. The simulation results are the average value of 30 simulations. The wireless network communication protocol used is IEEE 802.11, and each car is equipped with 2 antennas. The network bandwidth of the BS is 100 Mbps. The movement mode of each vehicle is the random way point. The data packet size is 1500 bytes. The probability of PU cars accessing the network is 0.3 to 0.8. The higher the probability, the lower the time and chance that SU can use. The number of spectrums for partial scanning is $\alpha = 3$, and the number of rounds for scanning is $\beta = 2$. The scanning spectrum adopts the passive handover mode.

**Table 1.** Simulation parameters.

| Parameter | Value |
|---|---|
| Simulation area | 2000 × 2000 (m$^2$) |
| Number of PU (PU$_{number}$) | 2, 4, 6, 8, 10 |
| Probability of PU car access network (P$_{PUaccess}$) | 0.3~0.8 |
| Number of SU (SU$_{number}$) | 2, 4, 6, 8, 10 |
| Number of BS | 10 |
| Transmission range of BS | 1000 (m) |
| Mobility model | Random way point |
| Moving speed of vehicle | 5~20 (m/s) |
| Bandwidth | 100 (Mbps) |
| Simulation time | 600 (sec) |
| Number of antenna per vehicle | 2 |
| Transmission range of vehicle | 100 (m) |
| Packet size | 1500 bytes |

### 4.3. Performance Results

(1)  Spectrum utilization

The spectrum usage is higher, the overall network performance is higher. Figures 13–15 show the performance result of spectrum utilization under different parameters. We can see that the cooperation-assisted mechanism can improve the spectrum utilization. Moreover, the partial scan scheme can reduce the scan time and increase the transmission time. Therefore, the performance of the partial scan scheme is better than the full scan scheme. In Figure 14,

when the moving speed of the vehicle increases, the SU must constantly switch BSs and then scan the spectrum, resulting in a significant drop in the overall spectrum utilization. In Figure 15, the higher the probability of the PU car accessing the network, it means that the PU itself is already using the spectrum, so the overall spectrum utilization rate is not too low. The SU can only access the spectrum through fragmented time, so the overall spectrum utilization rate performance improvement is limited. Interestingly, we can see the random access method. When the probability of PU cars accessing the network is low, the spectrum utilization rate will be higher than when the probability of PU cars accessing the network is high. This is because when the probability of the PU car accessing the network is high, the spectrum collisions of the SU will often occur, which will affect the performance of the PU. Therefore, when the probability of the PU car accessing the network is high, we suggest doing a spectrum scanning procedure so as not to cause performance inversion.

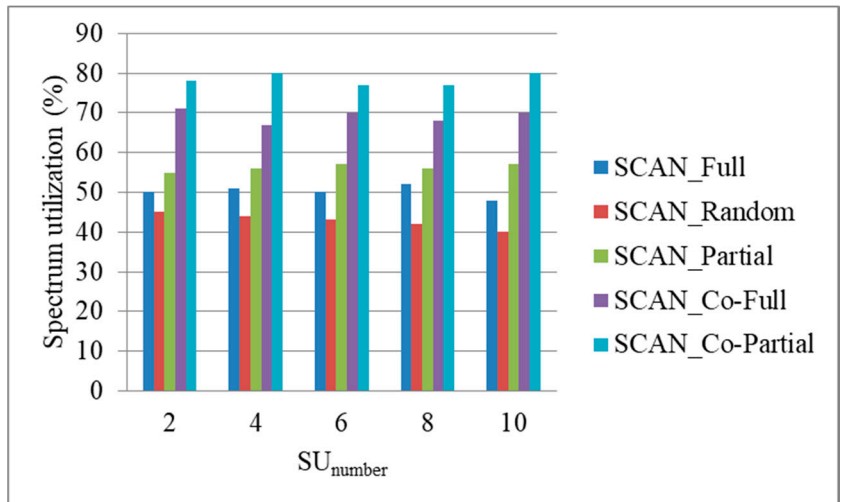

**Figure 13.** Spectrum utilization ratio under different number of SU cars ($PU_{number}$ = 10, $P_{PUaccess}$ = 0.5, V = 10).

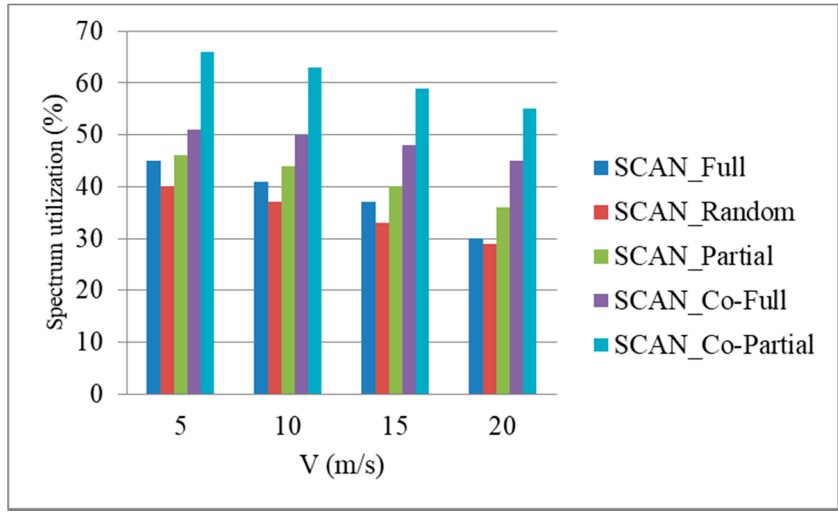

**Figure 14.** Spectrum utilization ratio under different moving speeds ($PU_{number}$ = 10, $P_{PUaccess}$ = 0.5, $SU_{number}$ = 10).

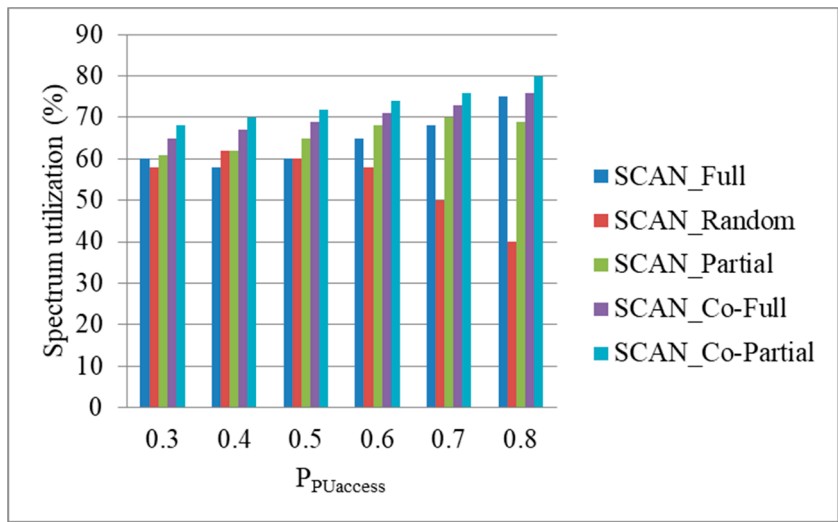

**Figure 15.** Spectrum utilization ratio under different primary user (PU) cars access network probabilities ($PU_{number}$ = 10, V = 10, $SU_{number}$ = 10).

(2)     Average scan time

In a fixed transmission cycle time, the spectrum scanning time is shorter, the interruption time of transmission is shorter and the increase of the time for transmitting data. From the results in Figures 16–18, the number of cars with different SUs has little effect on the average spectrum scanning time. However, as the moving speed increases and the probability of the PU vehicle access network increases, the average spectrum scanning time will increase proportionally. The scanning time of the random selection scheme is zero since there is no scanning behavior in this way. However, we can observe from other simulation results that although random access retains the most transmission time, the overall performance is not good. After excluding the randomly selection scheme, our proposed method has better results regardless of the number of SU cars, different moving speed, and the different probabilities of the PU cars accessing the network. This is because our method is to shorten the spectrum scanning time through partial scanning; then the SU obtains the spectrum usage of the new BS in advance through the cooperative assistance mechanism.

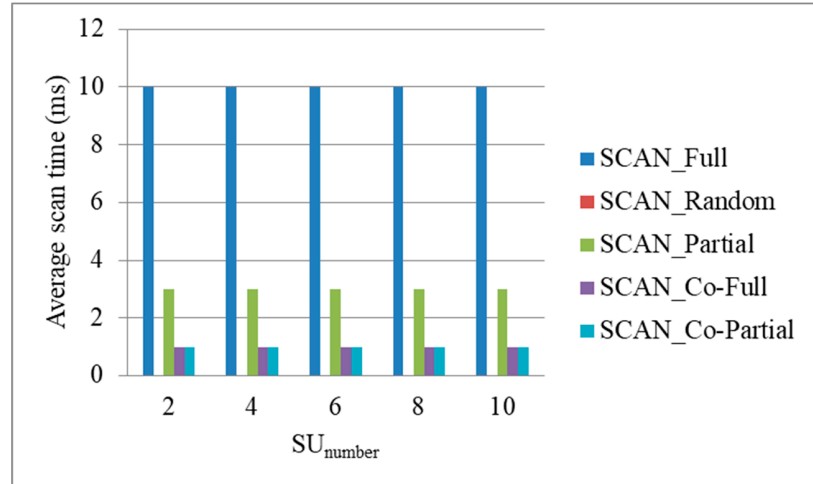

**Figure 16.** Average scan time under different number of SU cars ($PU_{number}$ = 10, $P_{PUaccess}$ = 0.5, V = 10).

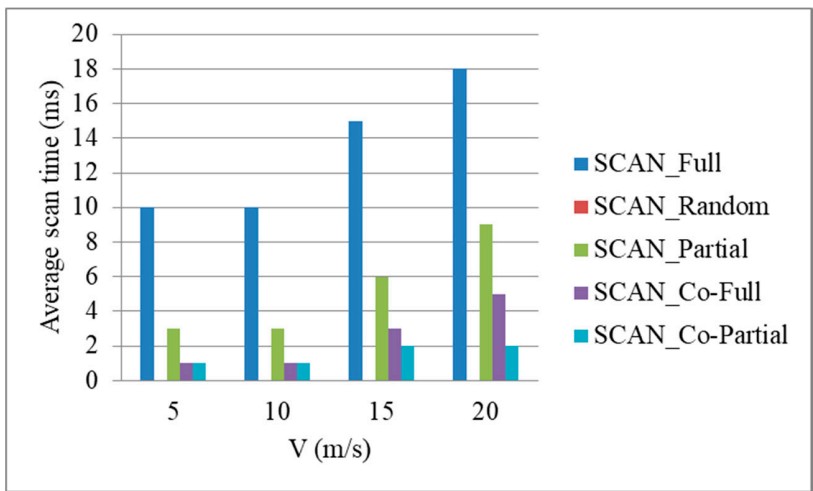

**Figure 17.** Average scan time under different moving speeds ($PU_{number}$ = 10, $P_{PUaccess}$ = 0.5, $SU_{number}$ = 10).

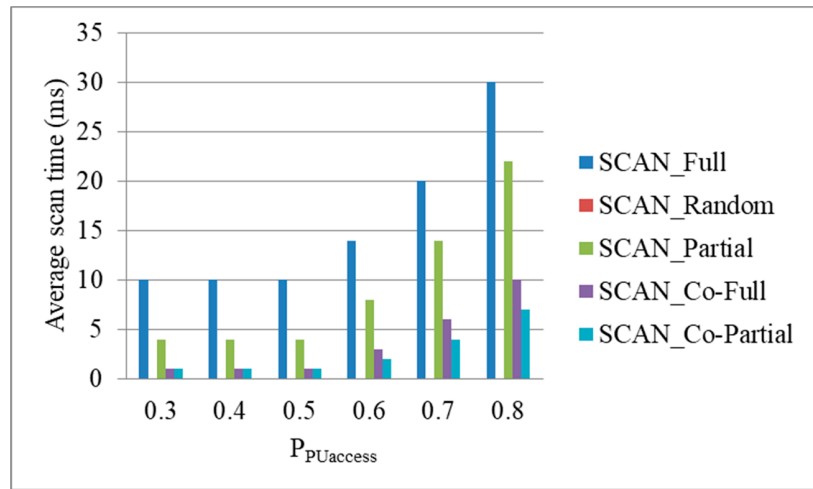

**Figure 18.** Average scan time under different PU cars access network probabilities ($PU_{number}$ = 10, V = 10, $SU_{number}$ = 10).

(3) Total number of handovers

The handover procedure is performed when the BS switches, the PU suddenly occupies the spectrum, or there is a transmission collision of the SU. If the spectrum handover is frequent, it means that the stability of the spectrum scanning mechanism is not good. From the results in Figures 19–21, we find that the more the numbers of SUs, the more handovers occur. The reason is that the SU competes for the spectrum or the transmission collision. Our cooperative assistance method has a lower total number of handovers because the passive handover mode is adopted. In addition, when switching to a new BS, the spectrum usage status of the new BS can be obtained in advance from the cooperative SU. Then the most suitable spectrum for handover is chosen. The random access method has the worst performance, because this method does not scan the spectrum, the access is performed directly, so collisions often occur, which leads to a rapid increase in the total number of handovers. When the moving speed of SU is faster, the total number of handovers will also increase. This is because SU will constantly switch BSs, so the SU is forced to do handover behavior. Our method has the lowest total number of deliveries because we consider the stability factor. When the PU often accesses the network, the total number of handovers will increase. This is because the SU may often transmit the data, and the PU will use the spectrum, which causes the SU to be forced to switch to other spectrums.

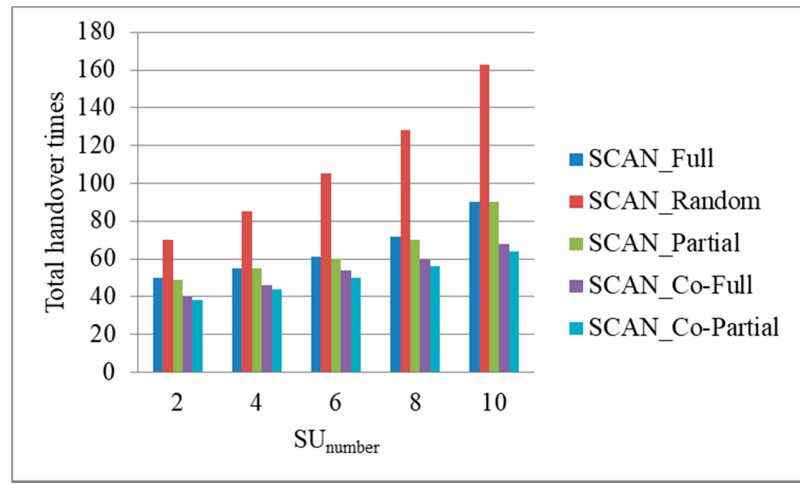

**Figure 19.** The total number of spectrum handovers under different number of SU cars ($PU_{number}$ = 10, $P_{PUaccess}$ = 0.5, V = 10).

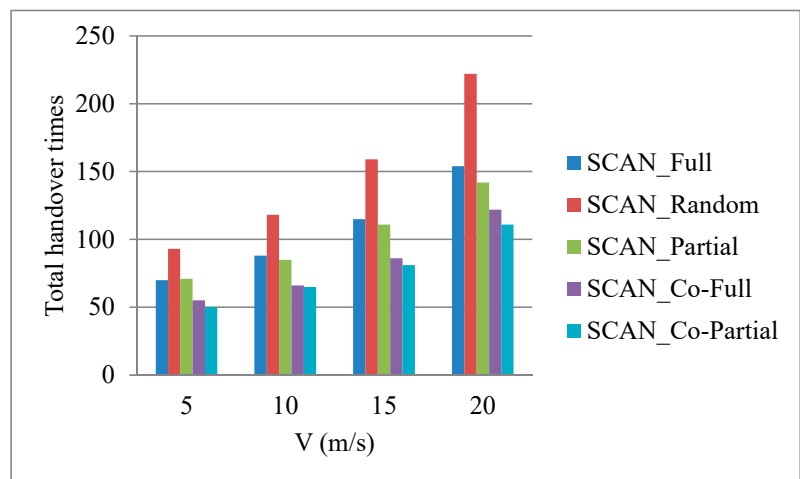

**Figure 20.** The total number of spectrum handovers under different moving speeds ($PU_{number}$=10, $P_{PUaccess}$=0.5, $SU_{number}$=10).

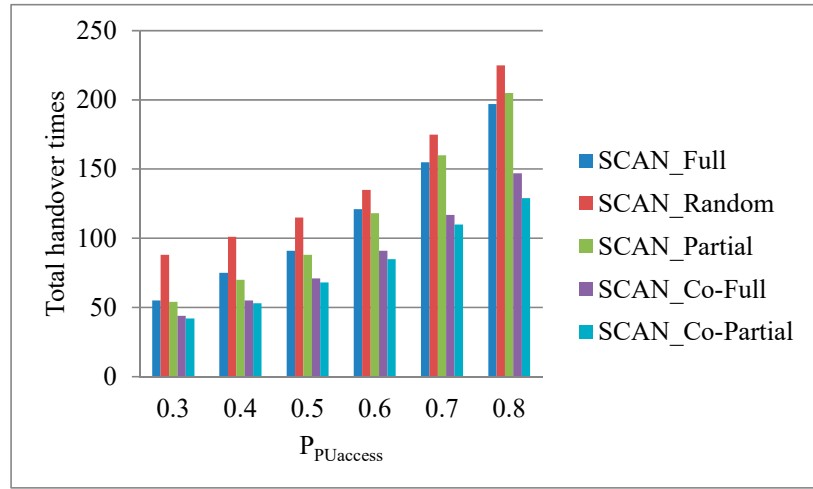

**Figure 21.** The total number of spectrum handovers under different PU cars access network probabilities ($PU_{number}$ = 10, V = 10, $SU_{number}$ = 10).

## 5. Conclusions

The spectrum handover is an important issue in the vehicular network. Therefore, this paper proposes a cooperation-assisted spectrum handover (CASH) scheme, which includes V2V cooperation, spectrum pre-scanning, fast partial scanning, and a dynamic scanning period. Then we use the network simulator to evaluate our method. According to the simulation results, the CASH mechanism can indeed lessen the scanning time, reduce the total number of handovers, and also greatly improve the overall spectrum utilization.

In the future, we will consider the following two aspects. First, we will consider the carrier aggregation (CA) technology, so that the idle available spectrum can provide greater bandwidth to achieve higher transmission efficiency. The second is that we hope to introduce a machine learning or deep learning scheme to make the spectrum selection mechanism more intelligent, while considering the problem of spectrum interference.

**Funding:** This research was funded by the Ministry of Science and Technology, R.O.C., under grants MOST 107-2221-E-163-001-MY3.

**Data Availability Statement:** Data are available upon request.

**Conflicts of Interest:** The authors declare no conflict of interest.

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
