# Peer review of "Cooperation-Assisted Spectrum Handover Mechanism in Vehicular Ad Hoc Networks"

_electronics, doi:10.3390/electronics10060742_

Round 1

Reviewer 1 Report

The author presented a Cooperation-Assisted Spectrum Handover Mechanism in Vehicular Ad Hoc Networks.

The article is well written and presents an interesting approach to resolve the handover issues in vehicular ad hoc networks.

Minor spelling and grammatical errors must be removed from the manuscript before final processing for publication. 

The article is accepted in current form. 

Reviewer 2 Report

Manuscript Identification #: electronics-1147284-peer-review-v1

Title: CASH: Cooperation-Assisted Spectrum Handover Mechanism in Vehicular Ad Hoc Networks

The research paper is focused on a cooperation-assisted spectrum handover (CASH) scheme. In CASH, each vehicle uses cognitive radio technology to collect the surrounding spectrum information, computes the stability of spectrum, exchanges information by vehicle-to-vehicle collaboration technology. The authors say that the simulation results show that proposed method outperforms other existing schemes.

In my opinion, the paper can be interesting from both academic and practical points, but it still needs a revision.

  • I really have problem with organization of this manuscript. A good scientific writing explains: 1. The goals of your experiment 2. How you performed the experiment 3. The results you obtained

, but I cannot see them in the manuscript about the results authors obtained. For example, there are many figures in this manuscript, 21 figures. Each of these figures includes 2 or 3 sub-figures, which makes the total number a kind of around 40 figures. This makes the paper tedious. It is hard to follow the trend of the paper in this sense. Can authors please remove figures that are not necessary. Alternatively, they can make the size of figures smaller or marge them. Just please use them in a more effective manner.

I also want to know why we should have assumptions 1 and 2 in Page 4. They over-simplify the problem unless there will be a reason for having them.

  • Abstract is the face of a paper. I think the start of abstract is more fit for Introduction section. The abstract should be started from Line 10 "This paper proposes a cooperationassisted spectrum handover (CASH) scheme where ..." and then author detail it in a better way.
  • The use of abbreviation in title and keywords is not common unless it is a well-known word. Please remove abbreviations “CASH:” and “(VANET)” from title and keywords, respectively.

Spectrum handover vehicular ad hoc network (VANET) --> Spectrum handover vehicular ad hoc network  

  • Authors do not need "etc." if they have already said "such as". Either one says that the list is not complete. Both makes it redundant. I can see this error in some sentences. For example, see

Page 4: such as inter-cell interference coordination (ICIC), interference cancellation technology, etc.

  • Collision (and collision probability) during transmission is studied in Page 8. I think having a collision-free SU is an important issue. The paper does not reflect the generality of collision avoidance in different areas, especially robotics as it is the origin of collision avoidance studies. The papers [a,b] should be cited for interested readers to know relevant applications where collision avoidance is important in, for example, robotics. [a] Robot motion planning using adaptive hybrid sampling in probabilistic roadmaps, Electronics 2016, 5(2), 16 [b] A cross-entropy method for optimising robotic automated storage and retrieval systems International Journal of Production Research, Vol 56, No 19, 6450-6472
  • As a kind suggestion, authors are better to use a more professional email, not “speedboy@”. However, they can keep this email if they have no other email.
  • Other errors:

Page 20: this paper propose --> this paper proposes

Reviewer 3 Report

Dear Authors.

This paper is good paper (CASH: Cooperation-Assisted Spectrum Handover Mechanism in Vehicular Ad Hoc Networks). (Moderate English changes required). But, I decision reconsider after major revision.

Strength of this paper included:

-The topic and is interesting.

-Cooperation-assisted spectrum handover mechanism is new.

Weakness of this paper:
1. 
I recommend additional/rewrite "Introduction".

2. 
The information presented is not new.

3.
I recommend additional/rewrite "Abstract and contribution".

4.
I recommend additional/rewrite "Introduction".

5. Related work: Improve
Important aspect has been mentioned.

5.1. "A comprehensive survey on vehicular ad hoc network." Journal of network and computer applications 37 (2014): 380-392.
5.2. Power Aware Routing Protocol in Multimedia Ad-hoc Network Considering Hop Lifetime of Node. Journal of Multimedia Information System, 1(2), 101-110.

- What is different Vehicular Ad Hoc Networks and Ad Hoc Networks.
clearly

-Also, more to 10 new papers published from 2019~2021 by major publishers such as IEEE, ACM, Springer, Elsevier, MDPI, and Wiley.

6.
Contribution need supported by data and result.

7. 
Results need clearly.

8.
Scientific Soundness: Low.

9.
English: Moderate English changes required.

Round 2

Reviewer 2 Report

I have read the paper once more, with the focus on the highlighted parts. Fortunately, the paper is improved in the current revision. Its organization is better now, and concepts such as Collision are covered. 

The typo errors are also resolved. Accordingly, I think the paper can be accepted as it is. 

Reviewer 3 Report

Dear Authors.

The revision adequately address the concerns expressed in last review. 
So, I recommend that this revised manuscript can now be recommended for publication. (Accept in present form)